# Identification of Novel mRNA Isoforms Associated with Acute Heat Stress Response Using RNA Sequencing Data in Sprague Dawley Rats

**DOI:** 10.3390/biology11121740

**Published:** 2022-11-29

**Authors:** Jinhuan Dou, Abdul Sammad, Angela Cánovas, Flavio Schenkel, Tahir Usman, Maria Malane Magalhães Muniz, Kaijun Guo, Yachun Wang

**Affiliations:** 1Animal Science and Technology College, Beijing University of Agriculture, Beijing 102206, China; 2Key Laboratory of Animal Genetics, Breeding and Reproduction, MARA, National Engineering Laboratory of Animal Breeding, Beijing Engineering Technology Research Center of Raw Milk Quality and Safety Control, College of Animal Science and Technology, China Agricultural University, Beijing 100193, China; 3Centre for Genetic Improvement of Livestock, Department of Animal Biosciences, University of Guelph, Guelph, ON N1G 2W1, Canada; 4College of Veterinary Sciences and Animal Husbandry, Abdul Wali Khan University, Mardan 23200, Pakistan

**Keywords:** heat stress response, novel transcripts, RNA-sequencing, blood, liver, adrenal glands, rats

## Abstract

**Simple Summary:**

Global warming and events of heat waves even in temperate climatic zones signify the importance of the genetic makeup of heat stress. Our earlier acute heat stress exposure study (from 30 min to 120 min treatments) in rats led to the designation of aberrant differentially expressed genes (DEGs) and pathways through RNA-sequencing of the three most relevant tissues, i.e., blood, liver, and adrenal glands. However, the causes and mechanisms of differential expression of genes associated with heat stress are still unclear. Using the same RNA-sequencing data, this study identified the differentially expressed mRNA isoforms and narrowed down the most reliable DEG markers and molecular pathways that underlie the thermoregulatory mechanisms. The spatial-temporal differential expression pattern of heat stress-responsive differential mRNA isoforms may be ultimately used in marker-assisted selection for improved thermotolerance.

**Abstract:**

The molecular mechanisms underlying heat stress tolerance in animals to high temperatures remain unclear. This study identified the differentially expressed mRNA isoforms which narrowed down the most reliable DEG markers and molecular pathways that underlie the mechanisms of thermoregulation. This experiment was performed on Sprague Dawley rats housed at 22 °C (control group; CT), and three acute heat-stressed groups housed at 42 °C for 30 min (H30), 60 min (H60), and 120 min (H120). Earlier, we demonstrated that acute heat stress increased the rectal temperature of rats, caused abnormal changes in the blood biochemical parameters, as well as induced dramatic changes in the expression levels of genes through epigenetics and post-transcriptional regulation. Transcriptomic analysis using RNA-Sequencing (RNA-Seq) data obtained previously from blood (CT and H120), liver (CT, H30, H60, and H120), and adrenal glands (CT, H30, H60, and H120) was performed. The differentially expressed mRNA isoforms (DEIs) were identified and annotated by the CLC Genomics Workbench. Biological process and metabolic pathway analyses were performed using Gene Ontology (GO) and the Kyoto Encyclopedia of Genes and Genomes (KEGG) database. A total of 225, 5764, and 4988 DEIs in the blood, liver, and adrenal glands were observed. Furthermore, the number of novel differentially expressed transcript lengths with annotated genes and novel differentially expressed transcript with non-annotated genes were 136 and 8 in blood, 3549 and 120 in the liver, as well as 3078 and 220 in adrenal glands, respectively. About 35 genes were involved in the heat stress response, out of which, *Dnaja1*, *LOC680121*, *Chordc1*, *AABR07011951.1*, *Hsp90aa1*, *Hspa1b*, *Cdkn1a*, *Hmox1*, *Bag3*, and *Dnaja4* were commonly identified in the liver and adrenal glands, suggesting that these genes may regulate heat stress response through interactions between the liver and adrenal glands. In conclusion, this study would enhance our understanding of the complex underlying mechanisms of acute heat stress, and the identified mRNA isoforms and genes can be used as potential candidates for thermotolerance selection in mammals.

## 1. Introduction

As the global temperature increases, heat stress has become a big challenge for human health and animal survival. An assessment report released by the Intergovernmental Panel on Climate Change (IPCC) predicts that by 2100, the global average surface temperature will rise by 0.3–4.8 °C [1], suggesting that it is more urgent to explore the body’s adaptation mechanism to heat stress and develop strategies to resist the adverse effects of heat stress. Heat stress is a major stressor affecting the health of animals, and body homeostasis is disturbed while employing thermoregulation [2]. The heat stress response is carried out through the hypothalamic–pituitary–adrenal (HPA) axis, causing hormonal changes including an increase in adrenocorticotropic hormones [3,4]. Homeotherms, upon heat stress, accelerate the respiratory rate, and sweating ensues, among other physiological responses [5,6]. Additional changes induced by heat stress cause spatiotemporal temperature distribution in various tissues [7]. Moreover, being a public health concern [8], heat stress causes physiological and biochemical changes in the body of rodents [9] and livestock [10,11,12]. These changes end up in the morbidity and decline of production and reproduction [10,13]. The negative effects of heat stress are characterized by high oxidative stress [14], molecular and transcriptional changes [15,16], along with post-transcriptional [17,18] and epigenetic changes [19] at the cellular and tissue levels. However, there is limited information about the molecular mechanisms of heat stress response in mammals.

Heat stress is a complex regulatory process combining the neurohormonal, oxidative, and immune responses [20,21,22]. Next-generation sequencing technologies, such as RNA sequencing (RNA-Seq), allow the large-scale, rapid, and accurate identification of heat stress-responsive genes, contributing to elucidating the molecular mechanisms of heat stress more deeply [23,24,25,26,27]. Moreover, numerous studies have demonstrated that the activity of some heat stress-responsive genes is not only regulated by their post-translational modifications [28,29,30] but also affected by changes in mRNA expression. Alternative splicing profoundly plays a vital role in many physiological processes. A study has shown that 60% of disease-causing point mutations are induced by interference with normal mRNA splicing during transcription [31]. In addition, alternative splicing was found to be involved in the process of biotic and abiotic stress responses [32,33,34]. As early as 1994, Takechi et al. reported that alternatively spliced mRNA with 169 additional nucleotides in the 5′ noncoding region was found after heat shock in comparison with mRNA transcribed under non-heat shock conditions [35]. Ju et al. found that in heat-stressed Bama miniature pigs, the second exon of the *TLR4* gene was spliced and 167 bp shorter in the alternative splicing variant [36]. In heat-stressed Drosophila, Nobuhiro et al. identified three new isoforms of heat shock transcription factor mRNA, including *HSFb*, *HSFc*, and *HSFd*, and found that the ratio of *HSFb* increased upon heat stress [28]. Although numerous studies reveal hundreds or thousands of heat stress-related genes in animals [9,37,38,39,40], studies focusing on large-scale mRNA isoforms in heat stressed-animals are still scarce in the literature.

In our previous study, the mRNA expression profiles of the blood, liver, and adrenal glands of rats in the control group (CT, 22 ± 1 °C, relative humidity 50%) and three heat-stressed groups fed at 42 °C for 30 min (H30), 60 min (H60) and 120 min (H120) (relative humidity 50%) were investigated using RNA-Seq [9,17,41]. In this context, the main hypothesis of the current study is that heat stress may change the number of mRNA isoforms involved in the expression levels of transcripts that are transcribed by genes involved in thermotolerance mechanisms in different tissues under different heat stress durations. Therefore, the objective of the present study was to identify the novel mRNA isoforms differentially expressed in blood, liver, and adrenal gland tissues of rats under H30, H60, and H120 conditions using RNA-Seq data obtained previously [9,17]. In addition, the mRNA isoforms identified in this study were integrated into different functional analyses in order to identify active physiological pathways and transcripts involved in the process of the heat stress response.

## 2. Materials and Methods

### 2.1. Animals and Sample Collection

Twenty 8-week-old female Sprague Dawley (SD) rats (Beijing Vital River Laboratory, Animal Technology Co., Ltd., Beijing, China) weighing 205 ± 7.16 g were to be used as subjects. During the whole experiment, water and feed were provided ad libitum. Based on changes in rectal temperature and 17 biochemical indicators of blood housed at H30, H60, and H120 conditions [41], a heat-stressed rat model was established in which 48 samples including blood (n = 4 each in the CT and H120 groups), liver (n = 5 each in the CT, H30, H60, and H120 groups), and adrenal glands (n = 5 each in the CT, H30, H60, and H120 groups) were collected. The blood, liver, and adrenal gland tissues in the same treatment group were collected from the same rat. Briefly, the experimental rats were anesthetized with an injection of 1%, 1.2 mL pentobarbital sodium (0.6 mL/100 g body weight), sacrificed, and quickly dissected by sterile surgical scissors and forceps (Shinva Medical Instrument Co. Ltd., Zibo, China) [42]. About 4 mL of blood was collected using Vacutainer^®^ tubes (Becton Dickinson, Plymouth, UK) containing Ethylenediaminetetraacetic acid (EDTA) from each rat and immediately placed on ice. Then, the peripheral blood mononuclear cells (PBMC) were collected by RNase-free spear via centrifugation (10 min, 3000 rpm) in a 2 mL centrifuge tube containing 1 mL Trizol (Invitrogen 15596018, Thermo Fisher Scientific Inc., Waltham, MA, USA) and stored at −80 °C for RNA extraction. The liver and adrenal glands were collected, washed in ice-cold phosphate-buffered solution, and snap-frozen immediately in liquid nitrogen.

### 2.2. RNA Isolation and Library Construction

The RNA reagent (HUAYUEYANG Biotechnology Co. Ltd., Beijing, China) was used to extract the total RNA from 48 samples of 20 SD rats according to the manufacturer’s protocol. The purity of the RNA samples was assessed at A260/230 nm and A260/280 nm ratios using the NanoDrop 2100 (Thermo Fisher Scientific Inc., Waltham, MA, USA). The integrity of the RNA (RIN) was determined using an RNA Nano 6000 Assay Kit of the Agilent Bioanalyzer 2100 system (Agilent Technologies, CA, USA), and the RNA concentration and genomic DNA contamination were determined using a Qubit^®^ 2.0 Fluorometer (Thermo Fisher Scientific Inc., Waltham, USA). The OD_260_/OD_280_ ratio varied from 1.8 to 2.0 for all samples, and RIN values > 7.0, indicating good RNA quality [43].

A total of 3 µg RNA from each sample was used to prepare the cDNA libraries using the NEBNext^®^ UltraTM Directional RNA Library Prep Kit from Illumina^®^ (NEB, San Diego, CA, USA), following the manufacturer’s recommendation. Paired-end (2 × 150 pb) reads of samples in the CT and H120 groups were sequenced using a HiSeq 2000 sequencer (Illumina, San Diego, CA, USA) in a previous study [9], and the reads of H30 and H60 groups were sequenced using the HiSeq 2500 sequencer (Illumina, San Diego, CA, USA) with 2 × 150 pb reads paired-end [17]. Both of the RNA-Seq protocols were performed in the Novogene Technology Co., Ltd., located in Tianjin of China.

### 2.3. Sequence Assembly and Quantification

All the transcriptomic analyses of the reads generated in CT, H30, H60, and H120 were performed using the CLC Genomics Workbench software 12.0 (CLC Bio, Aarhus, Denmark). Quality control analyses were performed following the parameters described in Cánovas et al. [44]. The paired-end sequenced reads were mapped to the annotated reference genome Rattus norvegicus 6.0 (rn6, ftp://ftp.ensembl.org/pub/release-105/genbank/rattus_norvegicus/, accessed on 11 February 2022) using the “Large Gap Read Mapping” tool. Briefly, this tool allows to map the RNA-Seq reads that span introns without requiring prior transcript annotations, which helps to conduct the transcript discovery analysis [45].

The transcript discovery analysis was performed using the “Transcript Discovery” tool implemented in the CLC Genomics Workbench environment (CLC Bio, Aarhus, Denmark) [45,46], which takes large gap read mapping tracks, uses gene and transcript annotations as an input, and produces optimized gene and transcript track annotations as outputs. Therefore, for each gene region, there was a set of transcript annotations that can explain the observed exons and splice sites in this region (data files available but not attached here to this manuscript) [46,47]. Then, the sequences of the samples were mapped to the created tracks (genes and transcripts tracks) using the rat reference genome as a map via the “RNA-Sequencing analysis tool” implemented in the CLC Genomics Workbench. The reads per kilobase per million mapped reads (RPKM) were used to quantify the expression levels for different transcripts across RNA-Seq libraries and were transformed by log10 [48,49]. Transcripts with RPKM ≥ 0.2 were defined as expressed. The expression levels of transcripts identified in each sample were classified as lowly expressed, moderately expressed, and highly expressed according to the criteria of RPKM ≤ 50, 50 < RPKM < 500, and RPKM ≥ 500 [44]. Principal component analysis (PCA, an unsupervised machine learning technique that seeks to find principal components) was performed on the gene expression by using the prcomp R function. The pair-wise Pearson correlation coefficient (PCC, r) between any two of the four or five biological replicates within each treatment of each comparison was also calculated by correlation procedure in the R package (version 4.2.0, https://cran.rproject.org/src/base/R-3/, accessed on 22 April 2022), where the Pearson’s correlation significance was computed using the Hmisc procedure.

### 2.4. Identification of Differential mRNA Isoform Expression

The CLC Genomics Workbench software 12.0 was used to conduct the differential mRNA isoform expression analyses [45,50]. Therefore, a two-stage experiment was carried out, applying empirical statistical analysis that implements an “Exact Test” developed by [51] and incorporated into the edgeR [52]. The statistical test was performed for each set up experiment, using the original count values and two parameters for the estimation of the dispersion: 1—“total count filter cut-off > 5, which specifies which features should be considered when estimating the common dispersion component”. 2—estimate tag-wise dispersions, which allow a weighted combination of the tag-wise and common dispersion for each transcript [45]. In addition, the original expression values were transformed based on a logarithm (log10) and normalized using scaling normalization described by [53] to ensure that the samples are comparable and assumptions on the data for analysis are met. The isoforms in the blood that met the thresholds of *p* < 0.05, |fold change (FC)| ≥ 2, and in the liver and adrenal glands met with *p* ≤ 0.001, FDR ≤ 0.05, and |FC| ≥ 2 were identified as differentially expressed (DEIs) [54], respectively.

### 2.5. Functional Enrichment Analysis and Gene Annotation

The main category biological process (BP) was performed using gene ontology (GO) enrichment analysis [55]. The analysis was performed using the list of genes associated with the DEIs in each tissue under various heat stress conditions. Functional evidence of the relationship between the significant GO terms (FDR ≤ 0.05) and the target phenotypes (e.g., response to heat stress) was identified. The metabolic pathway analysis was performed by the Kyoto Encyclopedia of Genes and Genomes (KEGG, http://www.genome.jp/kegg, accessed on 30 April 2022) database. The non-redundant biological terms for large clusters of genes in functionally related groups network were analyzed by ClueGO [56] and visualized by Cytoscape 3.8.2 [57].

To annotate novel mRNA isoforms of unknown genes, their coding sequence was obtained from the Genome Data Viewer tool, which is implemented in the National Center of Biotechnology Information (NCBI) database (https://www.ncbi.nlm.nih.gov/genome/gdv/browser/genome/, accessed on 6 May 2022). Thereafter, the ‘Nucleotide BLAST’ method available on the Basic Local Alignment Search Tool (BLAST; https://blast.ncbi.nlm.nih.gov/Blast.cgi, accessed on 6 May 2022) was used to detect similarities between the nucleotide sequences from the novel mRNA isoforms and the nucleotide (nt) collection available in the database. The nt collection consists of GenBank, EMBL, DDBJ, PDB, and RefSeq sequence data, but it excludes EST, STS, GSS, WGS, TSA, patent sequences, phase 0, 1, and 2 HTGS sequences, and sequences longer than 100 Mb. The database is non-redundant; i.e., identical sequences are merged into one entry (but it keeps the accession, GI, title, and taxonomy information from each entry).

Draw Venn Diagram (http://bioinformatics.psb.ugent.be/webtools/Venn/, accessed on 10 May 2022) was used to analyze the common and specific genes identified under various heat stress conditions and in different tissues. Finally, we searched our previous list of differentially expressed genes [41] for the genes associated with the differentially expressed known transcripts and the novel transcript lengths, defined them as DEIDEGs, and summarized the common DEIDEGs involved in the process of thermoregulation.

## 3. Results and Discussion

### 3.1. Global Landscape of the Rat Transcriptome

A total of 45 billion reads of 150 bp paired-end RNA-Seq data were produced, corresponding to an average of ~66 million reads per sample (Appendix A). High-quality clean reads were mapped to the rn6 genome. On average, 54,042, 86,680, and 102,927 transcripts were identified in the blood, liver, and adrenal glands, mapping to 34,239, 37,043, and 38,689 genes, respectively. The PCA results between any treatment samples within each tissue showed that the samples were divided mainly by various treatments (Appendix A). Then, the pair-wise PCC analysis yielded 6 or 10 pair-wise r values per sample group. For example, the mean r value in the CT of the blood was 0.9978 (ranging from 0.9971–0.9996), and the mean r value in the H120 group was 0.9995 (ranging from 0.9989–0.9998) (Appendix A). The results of all the PCCs indicated a high level of measurement consistency among the biological replicates (Appendix A).

Differences in the numbers of transcripts expressed among tissues were observed in the current study. On average, 11,288 (20.89%) of 54,042, 38,890 (46.02%), of 86,680, and 48,798 (47.10%) of 102,927 transcripts were defined as expressed (RPKM ≥ 0.2) in the blood, liver, and adrenal gland tissues, respectively (Figure 1). Among all the identified transcripts, a large proportion of lesser expressed transcripts (0.2 ≤ RPKM < 50) were identified in the blood (20.51%), liver (ranging from 43.67–45.31%), and adrenal glands (44.73–48.82%), followed by moderately expressed transcripts and highly expressed transcripts (Figure 1).

### 3.2. Identification of Various Types of the Differentially Expressed mRNA Isoforms

An overview of the DEIs between the treatments in each tissue is shown in Table 1. A total of 225, 2072, 2086, 1606, 1130, 1994, and 1864 DEIs were detected in the CT vs. H120 of blood, CT vs. H30, CT vs. H60, and CT vs. H120 comparisons of the liver and adrenal glands, respectively. A larger number of DEIs was identified in the liver compared with the adrenal glands and blood. Over all of the samples, the total number of novel differentially expressed transcript lengths of the annotated genes (6763) in all tissues was 1.75 and 19.43 times higher than the known DEIs (3866) and novel differentially expressed transcripts of non-annotated genes (348).

#### 3.2.1. Differentially Expressed mRNA Isoforms Annotated in the Rat Genome (Rattus Norvegicus 6.0)

A total of 81 known mRNA isoforms were differentially expressed in the blood in the CT vs. H120 comparison; 70.37% of them were up-regulated, and 29.63% were down-regulated. The *Rpl11-201* (ribosomal protein L11) was the top one annotated DEI with the highest FC of 206.84 (*p* = 8.34 × 10^5^), followed by *Rpl36al-201* (ribosomal protein L36a-like) (*p* = 1.75 × 10^4^) (Appendix A). In proliferating cells, the ribosomal stress response can be triggered by both increased and decreased ribosomal biogenesis, accompanied by the activation of the p53 pathway [58]. *Rpl11* may suppress cell growth via p53-dependent and/or independent mechanisms by interacting with other proteins and RNAs [59]. There is no research related to the role of *Rpl11* in heat stress. In addition, previous studies have reported that the transcription-independent p53 can trigger heat stress-induced apoptosis [60]. Therefore, it is necessary to further study whether *Rpl11* regulates cell activity during heat stress.

In the liver (Appendix A), an average of 698 known DEIs were detected in each comparison, and about 69.12% (482) of them were significantly down-regulated (*p* ≤ 0.001, FDR ≤ 0.05 and |FC| ≥ 2). Among all the comparisons of the liver, the most significantly changed known DEI was *Dbi-202* (diazepam-binding inhibitor, acyl-CoA-binding protein), with FCs of −6671.54 (FDR = 8.37 × 10^87^), −4386.08 (FDR = 1.54 × 10^85^), and −4386.08 (FDR = 1.54 × 10^85^) in CT vs. H30, CT vs. H60, and CT vs. H120 comparisons, respectively. The *Dbi* gene plays a role in the regulation of mitochondrial steroidogenesis and acyl-CoA metabolism. Previous studies have indicated that the DBI-related peptide can protect neurons and astrocytes from oxidative stress-induced apoptosis through its metabotropic receptor [61]. It is well known that oxidative stress occurs as a consequence of the imbalance between antioxidant defense and reactive oxygen species (ROS) production, and heat stress can stimulate the production of ROS [20], which suggests that *Dbi* may play an indirect role in the heat stress response. The Venn diagram analysis of all DEIs identified in three comparisons of the liver has shown that 97 isoforms were shared among all comparisons, including 83 significantly up-regulated isoforms and 14 significantly down-regulated isoforms (Figure 2A).

In adrenal glands, a total of 1690 isoforms were differentially expressed; 70.23%, 71.08%, and 65.83% of them were significantly up-regulated, which were 2.36, 2.46, and 1.93 times higher than down-regulated isoforms identified in CT vs. H30, CT vs. H60, and CT vs. H120 comparisons, respectively (Table 1). The *Rps15a-201* (ribosomal protein S15a) was the top one isoform that was differentially expressed (FDR = 2.43 × 10^60^) when H30 was compared to CT, with an FC of 3225.49 (Appendix A). The primary structure of *Rps15a* was firstly investigated in 1994, and it was shown that *Rps15a* has 129 amino acids [62]. Previous studies reported that *Rps15a* is engaged in the metabolism of RNA and proteins, rRNA processing, and translation [63,64,65]. In the comparison of CT vs. H60 (Appendix A), the most significantly changed isoform was *Hspe1-201* (heat shock protein family E (Hsp10) member 1), which is a stress-induced mitochondrial matrix protein and molecular chaperone [66]. However, no study was conducted on the function of *Hspe1* during the heat stress response. The *Giot1* (gonadotropin inducible ovarian transcription factor 1) was the top one significantly changed isoform in CT vs. H120 which was up-regulated (FC = 547.48, FDR = 2.96 × 10^30^). No shared annotated DEIs were detected among the three comparisons in the adrenal glands (Figure 2B).

#### 3.2.2. Differentially Expressed Novel Transcript Lengths for Genes Annotated in the Rat Genome Reference (Rattus Norvegicus 6.0)

In total, 60.44% (136), 61.57% (3549), and 61.71% (3078) of all DEIs in the blood (225), liver (5764), and adrenal glands (4988) were novel differentially expressed transcript lengths of annotated genes (Table 1). In the blood, transcript (ENSRNOG00000032844.4) with a 1198 bp length annotated to the known gene *RT1-Da* (RT1 class II, locus Da) was changed the most (FC = −115.88) and was significantly down-regulated at H120 (Appendix A). Furthermore, a large number of novel transcript lengths annotated for known genes that were identified as differentially expressed in blood were involved in the immune response, such as *RT1-Bb* (RT1 class II, locus Bb), *RT1-Da*, and *Crip1* (cysteine-rich protein 1).

In the liver, the novel transcript lengths of the annotated genes were found to be down-regulated (72.33% of 3549 in three comparisons), and 187 novel transcript lengths of the annotated known genes were shared among the CT vs. H30, CT vs. H60, and CT vs. H120 comparisons (Appendix A). Moreover, several shared genes were related to the response to the fatty acid metabolic process [such as *Hacl1* (2-hydroxyacyl-CoA lyase 1), *Acaca* (acetyl-CoA carboxylase alpha), *Fads1* (fatty acid desaturase 1), etc.]; the response to cold [such as *Gk* (glycerol kinase), *Hspd1* (heat shock protein family D member 1) and *Vegfa* (vascular endothelial growth factor A)]; the response to oxidative stress [such as *Abcb11* (ATP binding cassette subfamily B member 11), *Btg1* (BTG anti-proliferation factor 1), *Atrn* (attractin), *Gclc* (glutamate-cysteine ligase, catalytic subunit), *Ppif* (peptidylprolyl isomerase F), *Rcan1* (regulator of calcineurin 1), *Rps3* (ribosomal protein S3), and cell motility [such as *Slc9a3r1* (SLC9A3 regulator 1) and *Ctnna1* (catenin alpha 1)].

In adrenal glands (Appendix A), a total of 3078 novel transcript lengths annotated for known genes were differentially expressed depending on the thresholds of *p* ≤ 0.001, FDR ≤ 0.05, and |FC| ≥ 2, and 156 genes were commonly identified in three comparisons, including 63 genes that were up-regulated and 93 genes that were down-regulated when heat stress occurred. We also found that most shared genes were engaged in the response to hormones, the response to heat, and metabolic processes, such as *Btg1* (BTG anti-proliferation factor 1), *Lrp5* (LDL receptor-related protein 5), *Cyp11b3* (cytochrome P450, family 11, subfamily b, polypeptide 3), *Nr4a3* (nuclear receptor subfamily 4, group A, member 3), *Dnaja1* (DnaJ heat shock protein family (Hsp40) member A1), *Dnaja4* (DnaJ heat shock protein family (Hsp40) member A4), *Hspd1* and *Hsf1* (heat shock transcription factor 1), etc.

#### 3.2.3. Annotation of Differentially Expressed Novel Transcript Lengths Associated with Non-Annotated Genes in the Rat Genome Reference (Rattus Norvegicus 6.0)

The differentially expressed novel transcripts associated with non-annotated genes are shown in Table 2, Table 3 and Table 4, for blood (Table 2), liver (Table 3), and adrenal glands (Table 4), respectively. In total, 6, 48, and 66 differentially expressed novel transcripts with non-annotated genes were detected in the blood, liver, and adrenal glands, respectively. Furthermore, it was observed that those transcripts presented lengths between 248–16045 bp. Only the sequences of novel mRNA isoforms that had 100% similarity to the nucleotides in the NCBI database were retained.

Among the differentially expressed novel transcripts in the blood, *Gene_338_1* and *Gene_786_1* were down-regulated and presented 100% similarity with *Calm2* and *Gypa* (Table 2). *Calm2* is a kind of Ca^2+^-binding protein; it can modulate cell survival, apoptosis [67], and autophagy [68] via different pathways. A study has proved that *Calm2* is involved in cell stress regulation, especially endoplasmic reticulum stress [69]. *Gypa* is a major erythrocyte membrane sialoglycoprotein with a potential contribution to the development of diabetic complications [70]. In the liver (Table 3), 27, 15, and 6 differentially expressed novel transcripts associated with non-annotated genes were annotated in the rat genome (*Rattus norvegicus 6.0)*, when H30, H60, and H120 were compared to the CT groups, respectively. The *Gene_541*, located at the genome position “3:163817141-16381790”, had the greatest degree of change among the H30 (FC = −46.82), H60 (FC = −44.03), and H120 (FC = −43.04) treatment groups. A total of 35 differentially expressed novel transcripts were obtained in the H120 conditions in the adrenal glands, which is 1.46 and 5 times more than those transcripts in the H60 and H30 conditions (Table 4). Under H120, the gene encoding the heat shock protein 86 (Hsp86), also known as *Hsp90aa1*, was annotated by *Gene_4216*. Numerous studies have revealed the *Hsp90aa1* gene can be significantly up-regulated in a tissue-specific and time-dependent manner after heat stress [71]. Moreover, our previous study also confirmed the important role of *HSP70AA1* in dairy cattle [72]. Thus, the *Hsp90aa1* plays a pivotal role in the heat stress response.

### 3.3. Functional Enrichment Analysis

In order to annotate the DEIs further, GO and metabolic pathway analyses were performed on all of the annotated genes associated with DEIs (Appendix A). In summary, 185 genes were obtained in the blood (CT vs. H120) and 3189 genes were annotated in the liver, including 178, 1672, and 1339 genes in the CT vs. H30, CT vs. H60, and CT vs. H120 comparisons. In addition, a total of 3876 genes were annotated in the adrenal glands when the three heat treatment groups were compared to the CT.

#### 3.3.1. GO Analysis

Based on FDR ≤ 0.05, 77 BP terms were found in the blood for the CT vs. H120 comparison, and most of them were related to the immune response, such as the regulation of adaptive immune response (GO:0002819) and the regulation of T cell-mediated immunity (GO:0002709) (Appendix A). Furthermore, 48 genes were significantly enriched in terms of the response to stress (GO: 0006950) (FDR = 1.91 × 10^2^), including *Fn1*, *Rpl11*, *Dnajb2*, etc., which also have the functions of the positive regulation of T cell migration, the negative regulation of the apoptotic process, as well as the negative regulation of I-kappaB kinase/NF-kappaB (NF-κB) signaling. Heat stress is well documented to have a negative impact on the immune system through humoral immune responses and cell-mediated responses [73,74]. In the current research, all of the 77 significantly enriched BPs are directly or indirectly related to the immune response (Appendix A), suggesting that blood regulates heat stress by activating the immune response.

In the liver, totals of 139, 693, and 646 significantly (FDR ≤ 0.05) BP terms were detected in the CT vs. H30, CT vs. H60, and CT vs. H120 comparisons (Appendix A), respectively. Eighty of them were shared in all comparisons in the liver. The most significantly enriched term was the small molecule metabolic process (GO: 0044281). There were 395 BP terms significantly enriched by genes that were in common in at least two comparisons. Among them, the process of the response to stress (GO: 0006950) was significantly enriched by 339 and 279 genes in the CT vs. H60 and CT vs. H120 comparisons (Appendix A); 24 and 20 genes of them were also significantly (FDR ≤ 0.05) enriched in the process of the response to heat. Venn diagram analysis indicated that 17 genes were commonly identified in the H60 and H120 groups (Figure 3A), and they also play critical roles (*p* ≤ 0.05) in the response to arsenic-containing substances (Figure 3C). In addition, the positive regulation of the cold-induced thermogenesis (GO: 0120162) term was significantly (FDR = 3.91 × 10^2^) clustered by 14 genes that were specifically identified in the CT vs. H120 comparison (Figure 3B), and the other functions of these 14 genes are shown in Figure 3D.

In adrenal glands, 114, 649, and 567 BP terms were significantly (FDR ≤ 0.05) detected in the CT vs. H30, CT vs. H60, and CT vs. H120 comparisons, respectively (Appendix A). Among them, 353 terms were commonly detected in these three comparisons (marked in green color in Appendix A). In addition, the significant terms (FDR ≤ 0.05) related to the response to heat (GO: 0009408, Figure 4A) and cold-induced thermogenesis (GO: 0106106, Figure 4B) were enriched, and six genes including *Igfbp7*, *Dnaja1*, *LOC680121* (similar to heat shock protein 8), *Bag3*, *Hsp90aa1*, and *Hspd1* were shared in the CT vs. H30, CT vs. H60, and CT vs. H120 comparisons (Figure 4A). There were 19, 19, and 17 genes of the H30, H60, and H120 groups clustered in the process of cold-induced thermogenesis, and 5 genes were commonly identified among the CT vs. H30, CT vs. H60, and CT vs. H120 comparisons (Figure 4B). Furthermore, the *Hsf1* and *Trpv2* genes were significantly enriched both in the response to heat- and cold-induced thermogenesis processes at H60 and H120 (Figure 4A,B). Further functional annotation analysis was performed on genes that engaged in response to heat (n = 30) and cold-induced thermogenesis (n = 36). The results show that genes clustered in response to heat also play significant (*p* ≤ 0.01) roles in protein folding and the positive regulation of reactive oxygen species metabolic processes (Figure 4C). Genes clustered in the cold-induced thermogenesis process were also significantly (*p* ≤ 0.01) involved in the regulation of transcription from the RNA polymerase H promoter in response to stress, sterol homeostasis, and the regulation of DNA-template transcription in response to stress (Figure 4D).

It is widely known that changes in gene expression may be triggered by thermal stress; in turn, the expression of genes may help cells to defend themselves against thermal stress. Some genes were both activated by heat and cold stress, such as the *HSP70* (*HSP72*), *HSP90*, *HSF-1*, and *TRP* channels [75]. A previous study showed that the trimerization of *HSF-1* was induced by cold shock at 4 °C, and then increased the expression of *HSP70* and *HSP90* by binding to the heat response element [76]. This study also reported that *HSF-1* not only mediated the expression of HSP during the period of heat stress but also after rewarming to 37 °C. The TRP channels are often activated by a variety of stressors (e.g., heat, cold, pH). *TRPV1-4* and *TRPM2* are activated by heat stress, and *TRPA1* and *TRPM8* are activated by cold stress [77]. In the present study, *HSF-1* and *Trpv2* were found to be involved in the process of the response to heat- and cold-induced thermogenesis at H60 and H120, suggesting that both *HSF-1* and *Trpv2* played important roles in thermoregulation, but further validation experiments are needed to confirm their functions. The *Trpv2* gene can be considered as a novel target for a subsequent heat stress study.

#### 3.3.2. Metabolic Pathway Analysis

Thirty-three metabolic pathways were significantly detected in the blood, which were divided into five categories, including the cellular process, the environment information process, human diseases, metabolism, and organismal systems (Appendix A). A total of 57.58% (n = 19) of the 33 pathways were related to human disease, containing cardiovascular disease, immune disease, and energy metabolism. The most significantly enriched pathway was viral myocarditis (FDR = 8.73 × 10^10^), which clustered 13 genes, including *RT1-Ba*, *Prf1*, *RT1-CE5*, *RT1-CE7*, *RT1-T24-1*, *RT1-Bb*, *RT1-Da*, *RT1-Db1*, *Actb*, *Actg1*, *RT1-CE4*, *RT1-CE10*, and *LOC108348139* (Appendix A). Kanda et al. proved that heat stress can aggravate viral myocarditis in mice by inducing viral infection [78].

In the liver, 31, 46, and 51 pathways were significantly enriched in the CT vs. H30, CT vs. H60, and CT vs. H120 comparisons, respectively (Appendix A). Eleven pathways were commonly shared in the three comparisons, including fatty acid metabolism (ko01212) and the PPAR signaling pathway (ko03320) (Appendix A). Fatty acids can regulate immune homeostasis by the activation of fatty acid receptors involved in inflammation and oxidative stress during heat stress [79,80]. Heat stress increases the expression levels of certain genes involved in promoted fatty acid metabolism, such as PPARγ, CCAAT-enhancer-binding protein α (CEBPα), and fatty acid synthase (FAS) [80]. Importantly, the PPAR signaling pathway activated by fatty acids regulates inflammatory responses and lipid metabolism [79,81,82]. The fatty acid metabolism-related genes and pathways identified in this study show the complexity of heat stress response and its possible implications for the welfare and production traits of livestock in summer. There were also 11 and 67 genes in the H30 and H60 groups significantly clustered in the thermogenesis pathway (ko04714), which help animals adapt to environmental changes. In adrenal glands, 47, 20, and 19 pathways were significantly detected by 1234, 1416, and 1226 genes in the CT vs. H30, CT vs. H60, and CT vs. H120 comparisons (Appendix A). Only two pathways, including protein processing in the endoplasmic reticulum (ko04141) and ferroptosis (ko04216), were commonly detected by genes in all three comparisons. The pathway of thermogenesis (ko04714) was detected by 38 and 35 genes in the H30 and H60 groups, respectively. (Appendix A).

### 3.4. Summary of Differential Isoform Corresponding Genes That Were also Differentially Expressed in Tissues

A total of 43, 786, and 1149 DEIDEGs were found in blood, liver, and adrenal gland tissues under heat stress, respectively (Table 5). Genes enriched in the heat stress process and thermogenesis are shown in Table 6. A total of 35 genes were enriched in the liver and adrenal glands, including 10 shared genes, such as *Dnaja1*, *LOC680121*, *Chordc1*, *AABR07011951.1*, *Hsp90aa1*, *Hspa1b*, *Cdkn1a*, *Hmox1*, *Bag3*, and *Dnaja4*, suggesting that these genes may regulate the heat stress response through crosstalk between the liver and adrenal glands. Combining with the expression profiles from the RNA-Seq and the validation results of the RT-qPCR experiments performed in our previous study [9], which demonstrated the high reliability of the RNA-Seq technology, we preferentially suggest that these 35 genes could be valuable candidates for underlying mechanisms to cope with heat stress.

Understanding mechanisms of heat stress response and discovering indicators to quantify the degree of heat stress are challenging for researchers [82,83]. Numerous studies have evidenced that gene expression analysis is a great strategy to identify positional and functional candidate genes associated with an organism’s response to heat stress [45]. Furthermore, the development of RNA-Seq technology has greatly improved the accuracy and comprehensiveness of gene expression analyses and accelerated the progress of heat stress-related research [84,85]. In the current study, RNA-Seq analysis was performed on three tissues (blood, liver, and adrenal glands) and four treatment groups (CT, H30, H60, and H120) and was not only able to identify the known mRNA isoforms annotated to known genes but also detected some novel transcripts associated with non-annotated genes. Our study also indicated that the expression of heat stress-responsive genes has spatial and temporal differences [17]. Moreover, the novel mRNA isoforms identified in this study would contribute to further studies, providing candidate mechanisms related to heat stress response, and to enriching the rat transcriptome map. If these novel mRNA isoforms are common to different mammalians, these regions could be used to identify universal markers that could be used for breeding for heat-resistant livestock mammals.

## 4. Conclusions

This research described the transcriptome profiles in the blood, liver, and adrenal glands of Sprague Dawley rats under various heat stress conditions, showing that the expression of heat stress-responsive genes in rats has spatiotemporal differences. In total, 35 genes were involved in the process of thermal stress regulation annotated by DEIs and were also differentially expressed in the liver and adrenal gland tissues, which could be preferentially considered as candidate markers for heat stress in rats. Furthermore, the identification of novel mRNA isoforms related to heat stress response allows for new insights into a better understanding of transcripts functionally playing important roles in heat stress, which may in the future be used to select for thermotolerance. The evident role of the inflammation–metabolism nexus appears to be central in the acute-heat-stressed rats through mRNA isoform identification, and hence this study indirectly emphasizes this intervention nexus for heat stress management. The magnitude of transcriptional changes uncovered in this study points towards the need for pathway-level biological investigation, along with further studies on the essential crosstalk between different biological pathways. Future GWAS studies focusing on the thermotolerance breeding of livestock can benefit from this study in the selection of genes, and a careful approach taking into account the spatiotemporal and post-transcriptional changes can be adopted.

## Figures and Tables

**Figure 1 biology-11-01740-f001:**
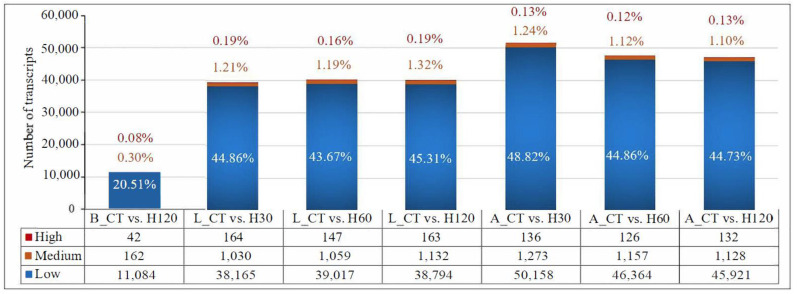
Statistics of the number of transcripts identified in blood, liver, and adrenal glands under various heat stress conditions. High, medium, and low mean the high, moderate, and low expressed transcripts. The statistic numbers marked in white, orange, and red means the ratio of transcripts with different expression levels to all transcripts identified in blood, liver, and adrenal glands under various heat stress conditions. B, L, and A mean blood, liver, and adrenal gland tissues, respectively. CT means rats housed at 22 ± 1 °C and relative 50% humidity; the H30, H60, and, H120 mean the rats exposed to 42 °C for 30 min, 60 min, and 120 min with a relative humidity of 50%.

**Figure 2 biology-11-01740-f002:**
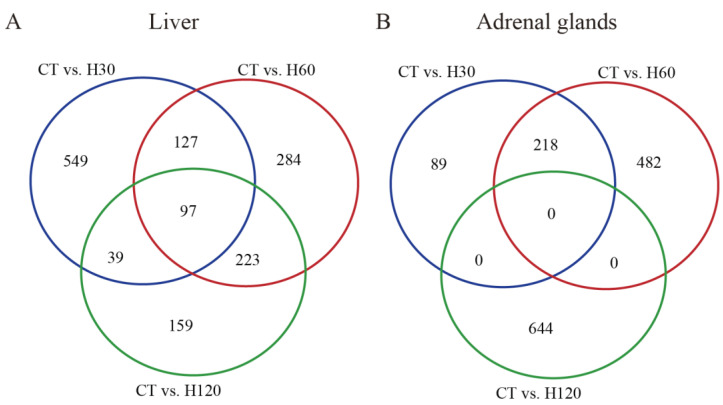
Venn diagrams of the known differentially expressed mRNA isoforms identified in all comparisons. (**A**) Venn diagram of the shared DEIs in the liver among three comparisons. (**B**) Venn diagram of the shared DEIs in adrenal glands among three comparisons. The blue, red, and green circles refer to the comparisons of CT vs. H30, CT vs. H60, and CT vs. H120, respectively.

**Figure 3 biology-11-01740-f003:**
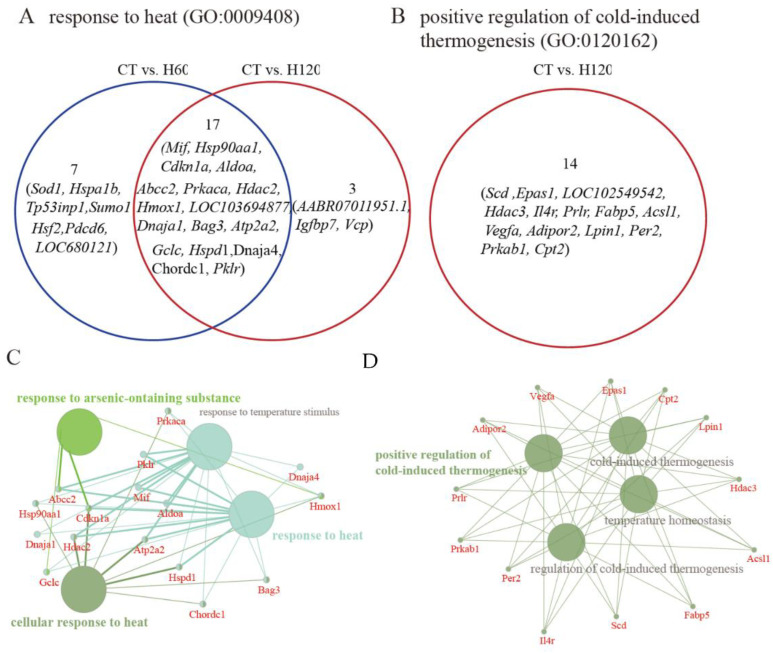
Genes of liver involved in the heat stress process and their functional annotations. (**A**) Venn diagram of genes that engaged in terms of response to heat (GO: 0009408). (**B**) Venn diagram of genes that engaged in terms of the positive regulation of cold-induced thermogenesis (GO: 0120162). (**C**) The functional annotation of genes that were significantly clustered in response to heat and were shared between the CT vs. H60 and CT vs. H120 comparisons. (**D**) The functional annotation of genes in the CT vs. H120 comparison of the liver that were significantly clustered in the positive regulation of cold-induced thermogenesis. Only the GO-BP terms with *p* ≤ 0.05 are shown in colored words.

**Figure 4 biology-11-01740-f004:**
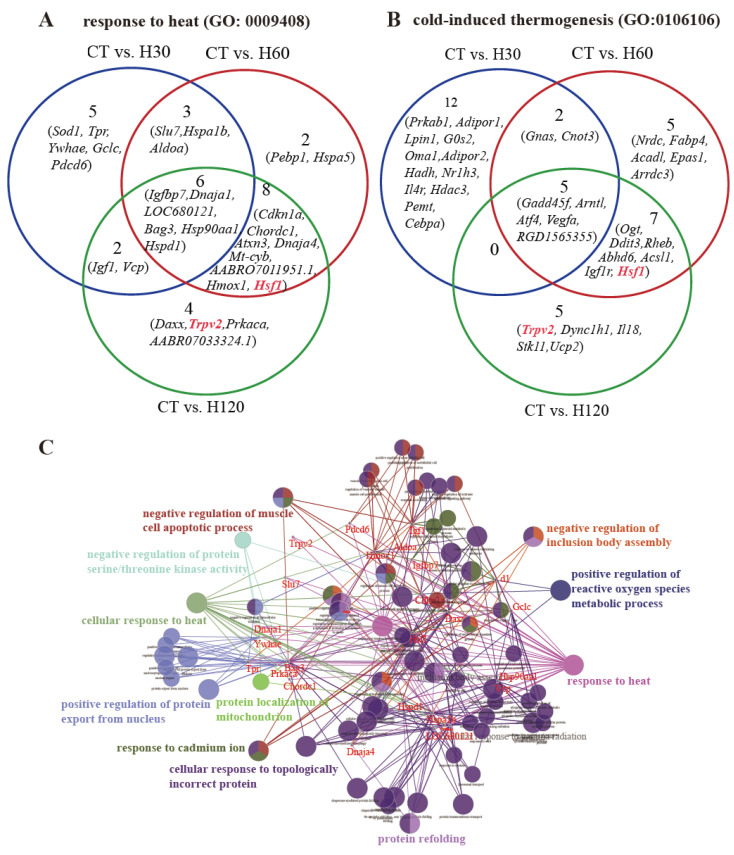
Analysis of genes in adrenal glands engaged in thermal regulation processes. (**A**) Venn diagram of the genes involved in process of response to heat (GO: 0009408). (**B**) Venn diagram of the genes involved in the process of cold-induced thermogenesis (GO: 0106106). (**C**) Functional annotation of genes enriched in the process of the response to heat. The GO-BP terms with *p* ≤ 0.01 were regarded as significant. (**D**) Functional annotation of genes enriched in the process of cold-induced thermogenesis. The GO-BP terms with *p* ≤ 0.01 were regarded as significant.

**Table 1 biology-11-01740-t001:** Summary of DEIs identified in blood, liver, and adrenal glands of rats under heat stress.

Comparisons	DEIs Annotated with Associated Known Genes			Novel Differentially Expressed Transcript Lengths of Annotated Genes			Novel Differentially Expressed Transcripts with Non-Annotated Associated Genes		
	Total	Up	Down	Total	Up	Down	Total	Up	Down
B_CT vs. H120	81	57	24	136	86	50	8	6	2
L_CT vs. H30	812	99	713	1214	413	801	46	16	30
L_CT vs. H60	731	332	399	1311	301	1010	44	14	30
L_CT vs. H120	552	216	336	1024	268	756	30	12	18
A_CT vs. H30	309	217	92	788	492	296	33	17	16
A_CT vs. H60	702	499	203	1208	709	499	84	57	27
A_CT vs. H120	679	447	232	1082	725	357	103	83	20

Note: CT means control group, rats housed at 22 ± 1 °C and relative humidity of 50%; H30, H60, and H120 mean 42 °C heat stress for 30 min (H30), 60 min (H60), and 120 min (H120) with a relative humidity of 50%; DEIs means differentially expressed mRNA isoforms; Up means the expression of the mRNA isoforms was up-regulated in the heat treatment group compared to the CT; Down means the expression of the mRNA isoforms was down-regulated in the heat treatment group compared to the CT. B, L, and A refer to blood, liver, and adrenal gland tissues.

**Table 2 biology-11-01740-t002:** Differentially expressed novel transcripts associated with non-annotated genes in blood.

Feature ID	Position	mRNA Length (bp)	*p*-Value	FC	E-Value	Identity (%)	Pred. Gene Accession	Predicted Gene
Gene_338_1	6:11067663-11069810	1225	3.97 × 10^2^	−2.52	0.00 × 10^0^	100.00	BC058485.1	Calm2
Gene_45_2	1:148454137-148457578	414	3.52 × 10^3^	9.19	8.49 × 10^−29^	100.00	BC091223.1	Mpp1
Gene_45_3	1:148454178-148457578	373	9.40 × 10^3^	2.86	8.49 × 10^−29^	100.00	BC091223.1	Mpp1
Gene_739_2	17:36393064-36395863	553	3.48 × 10^2^	6.23	0.00 × 10^0^	100.00	XR_598144.1	LOC103694081
Gene_750_1	17:85715378-85749024	833	1.18 × 10^2^	4.62	3.44 × 10^−121^	100.00	AB032899.1	Pip4k2a
Gene_786_1	19:31115483-31131085	1405	2.48 × 10^2^	−16.84	0.00 × 10^0^	100.00	FQ225003.1	Gypa

Notes: FC = fold change; Identify = the highest percent identity for a set of aligned segments to the same subject sequence; Pred. gene accession = accession number of predicted genes in the NCBI database.

**Table 3 biology-11-01740-t003:** Differentially expressed novel transcripts associated with non-annotated genes in the liver.

Feature ID	Position	mRNA Length (bp)	*p*-Value	FC	E-Value	Identity (%)	Pred. Gene Accession	Predicted Gene
Gene_218	1:227110795-227113948	1023	8.98 × 10^4^	−2.15	2.23 × 10^14^	100.00	FQ229791.1	TL0ADA46YK21
Gene_544	3:165405586-165412314	836	2.18 × 10^9^	−2.32	4.86 × 10^4^	100.00	AC091536.2	clone RP31-547A5
Gene_668	4:180315560-180317130	810	9.84 × 10^4^	−15.11	2.40 × 10^6^	100.00	AC091503.2	clone RP31-446A15
Gene_360	2:225812754-225827011	291	2.70 × 10^4^	−7.31	7.98 × 10^5^	100.00	AC087067.3	RP31-40H10
Gene_45	1:77219372-77222802	3430	6.65 × 10^4^	5.42	6.86 × 10^5^	100.00	AC091514.2	RP31-223K12
Gene_277	2:34923108-34940410	382	3.54 × 10^12^	−6.06	1.00 × 10^3^	100.00	AC091514.2	clone RP31-223K12
Gene_1132	8:127840013-127845706	1258	7.93 × 10^4^	−5.85	6.83 × 10^7^	100.00	AC087722.2	RP31-198L13
Gene_1132	8:127837857-127845709	2438	3.37 × 10^4^	−3.22	9.43 × 10^7^	100.00	AC087722.2	clone RP31-198L13
Gene_2893	11:81514108-81516760	278	2.02 × 10^4^	−23.14	5.29 × 10^5^	100.00	AC087722.2	clone RP31-198L13
Gene_1241	10:14087347-14088001	419	3.00 × 10^4^	−5.45	5.83 × 10^9^	100.00	AC087262.2	RP31-151E23
Gene_1814	16:70876536-70896702	1230	1.00 × 10^4^	−9.08	5.22 × 10^8^	100.00	AC087262.2	clone RP31-151E23
Gene_69	1:90859863-90889196	226	2.87 × 10^4^	−2.84	1.64 × 10^4^	100.00	AC079389.2	RP31-263K14
Gene_1894	18:3426564-3437474	1205	5.46 × 10^4^	−2.27	7.89 × 10^4^	100.00	AC079389.2	clone RP31-263K14
Gene_812	6:60958385-61055343	434	6.73 × 10^4^	−2.66	2.39 × 10^42^	100.00	AC079378.2	clone RP31-7L11
Gene_262	2:22812846-22819691	524	8.38 × 10^4^	4.76	4.74 × 10^34^	100.00	AC079378.2	clone RP31-7L11
Gene_661	4:175847044-175882304	6963	7.15 × 10^13^	7.19	9.13 × 10^8^	100.00	AC079378.2	clone RP31-7L11
Gene_1000	7:127445500-127447242	1742	7.06 × 10^4^	6.96	0.00	100.00	XM_017595283.1	LOC678774
Gene_2063	KL568103.1:3964-4554	404	1.18 × 10^7^	−5.11	1.13 × 10^4^	100.00	AB002169.1	RT1
Gene_102	1:137413556-137422477	8921	6.05 × 10^5^	5.79	1.39 × 10^4^	100.00	AC105654.4	Renin BAC CH230-201P14
Gene_102	1:137413555-137427301	12812	8.50 × 10^5^	11.95	2.14 × 10^4^	100.00	AC105654.4	Renin BAC CH230-201P14
Gene_2027	20:5353181-5354060	840	4.67 × 10^5^	−4.90	7.89 × 10^4^	100.00	AJ314857.1	Atp6G
Gene_886	7:3051669-3053895	1569	1.79 × 10^10^	−5.63	1.59 × 10^4^	100.00	AB218617.1	clone CH230-65K18
Gene_662	4:176559544-176565091	3618	4.23 × 10^4^	7.85	6.56 × 10^17^	100.00	AC092530.35	clone rp32-28p17
Gene_245	1:266346460-266355969	2762	9.70 × 10^9^	−5.97	1.48 × 10^5^	100.00	AC094697.7	BAC CH230-5F4
Gene_2086	X:74324189-74329900	5711	1.34 × 10^16^	4.71	4.12 × 10^4^	100.00	AC107611.6	10 BAC CH230-195I24
Gene_2086	X:74324189-74330435	1143	4.22 × 10^7^	3.32	4.51 × 10^4^	100.00	AC107611.6	10 BAC CH230-195I24
Gene_541	3:163817141-163817905	296	7.88 × 10^4^	−44.03	0.00	100.00	FO181541.11	clone bRB-233C6
Gene_859	6:125861248_125870177	363	5.82 × 10^10^	−2.42	6.45 × 10^4^	100.00	AC091503.2	clone RP31-446A15
Gene_859	6:125861248_125870115	298	1.27 × 10^9^	−2.65	6.41 × 10^4^	100.00	AC091503.2	clone RP31-446A15
Gene_409	3:23268250_23272886	3211	8.75 × 10^4^	−7.90	4.08 × 10^43^	100.00	AC091503.2	clone RP31-446A15
Gene_360	2:225812754_225827011	291	3.59 × 10^4^	-6.17	7.98 × 10^5^	100.00	AC087067.3	clone RP31-40H10
Gene_277	2:34923108_34940410	382	7.82 × 10^11^	−5.16	1.00 × 10^3^	100.00	AC091514.2	clone RP31-223K12
Gene_3099	11:81514105_81516759	280	4.30 × 10^4^	−12.03	5.29 × 10^5^	100.00	AC087722.2	clone RP31-198L13
Gene_886	7:3051672_3053895	1566	6.61 × 10^11^	−5.93	1.59 × 10^4^	100.00	AB218617.1	clone CH230-65K18
Gene_2215	1:187792023_187853814	9668	3.79 × 10^4^	14.36	3.27 × 10^44^	100.00	AB218617.1	clone CH230-65K18
Gene_679	5:33659016_33696506	464	8.46 × 10^5^	8.85	9.70 × 10^8^	100.00	AC094583.7	BAC CH230-4M16
Gene_738	5:150489343_150492182	479	1.52 × 10^4^	2.12	3.22 × 10^42^	100.00	AC094583.7	BAC CH230-4M16
Gene_797	6:21900746_21915819	2045	2.64 × 10^6^	−6.86	3.03 × 10^4^	100.00	AC096165.10	BAC CH230-11H2
Gene_838	6:102105282_102134302	541	5.13 × 10^6^	−4.44	4.52 × 10^5^	100.00	AB049248.2	Atrn gene
Gene_413	3:36034086_36035501	334	2.21 × 10^4^	−4.57	7.77 × 10^6^	100.00	AC091353.6	BAC CH230-1B20
Gene_45	1:77213725_77228134	14409	1.75 × 10^4^	10.13	2.90 × 10^4^	100.00	AC096051.7	BAC CH230-21G1
Gene_541	3:163817141_163817905	296	7.23 × 10^4^	−46.82	0.00	100.00	FO181541.11	clone bRB-233C6
Gene_738	5:150489343-150492182	479	2.02 × 10^5^	2.30	3.22 × 10^42^	100.00	AC094583.7	BAC CH230-4M16
Gene_44	1:77203585-77213633	1006	3.68 × 10^5^	4.39	1.21 × 10^6^	100.00	AC106169.5	BAC CH230-105D14
Gene_838	6:102105282-102134302	541	3.86 × 10^5^	−3.69	4.52 × 10^5^	100.00	AB049248.2	Atrn
Gene_886	7:3051668-3053625	1523	2.53 × 10^4^	−2.24	1.40 × 10^4^	100.00	AB218617.1	clone CH230-65K18
Gene_45	1:77214418-77228129	6759	3.84 × 10^4^	7.73	2.76 × 10^4^	100.00	AC096051.7	BAC CH230-21G1
Gene_541	3:163817141-163817905	296	8.31 × 10^4^	−43.04	0.00	100.00	FO181541.11	clone bRB-233C6

Notes: FC = fold change; Identify = the highest percent identity for a set of aligned segments to the same subject sequence; Pred. gene accession = accession number of predicted genes in the NCBI database. The blue, orange, and green backgrounds mean results generated in CT vs. H30, CT vs. H60, and CT vs. H120 comparisons.

**Table 4 biology-11-01740-t004:** Differentially expressed novel transcripts associated with non-annotated genes in adrenal glands.

Feature ID	Position	mRNA Length (bp)	*p*-Value	FC	E-Value	Identity (%)	Pred. Gene Accession	Predicted Gene
Gene_754	3:154869698-154872439	2741	8.27 × 10^7^	34.47	6.69 × 10^44^	100.00	AC079378.2	clone RP31-7L11
Gene_3637	5:157946908-157956567	1627	4.62 × 10^4^	16.08	1.14 × 10^16^	100.00	AC109542.6	BAC CH230-270O15
Gene_481	2:183542130-183582702	411	4.78 × 10^4^	−21.50	3.78 × 10^7^	100.00	AC087262.2	clone RP31-151E23
Gene_88	1:81050735-81058656	339	7.50 × 10^5^	−27.62	9.51 × 10^7^	100.00	AC095281.6	BAC CH230-10M16
Gene_3637	5:157939068-157956567	2706	4.61 × 10^4^	6.99	3.52 × 10^4^	100.00	AC105515.5	BAC CH230-13H11
Gene_3633	5:155905289-155912270	1803	1.83 × 10^5^	−32.28	5.04 × 10^4^	100.00	AB294577.1	chromosome 13q11-q12
Gene_3748	6:133793172-133801420	3252	2.34 × 10^4^	−2.36	5.96 × 10^4^	100.00	AC079389.2	clone RP31-263K14
Gene_916	4:157118179-157122026	1796	4.73 × 10^5^	−2.72	2.77 × 10^4^	100.00	AC241808.7	BAC RNECO-49K24
Gene_444	2:112717826-112723008	842	5.03 × 10^4^	10.76	3.71 × 10^9^	100.00	AC094963.9	BAC CH230-6L20
Gene_187	1:146659430-146706927	2531	7.03 × 10^4^	2.72	7.35 × 10^10^	100.00	AC087775.2	clone RP31-464J4
Gene_8	1:15748741-15760499	4456	1.67 × 10^4^	6.87	8.51 × 10^4^	100.00	AC087722.2	clone RP31-198L13
Gene_2499	15:51276012-51303786	248	1.39 × 10^4^	6.51	1.56 × 10^4^	100.00	AC087112.2	clone RP31-162L19
Gene_1808	10:16149054-16155114	4974	1.88 × 10^7^	24.50	3.38 × 10^5^	100.00	AC090529.2	clone RP31-160L19
Gene_1808	10:16149054-16155114	5049	5.93 × 10^6^	14.63	3.38 × 10^5^	100.00	AC090529.2	clone RP31-160L19
Gene_1808	10:16149054-16155114	5639	7.33 × 10^6^	7.73	3.38 × 10^5^	100.00	AC090529.2	clone RP31-160L19
Gene_1808	10:16149054-16155114	4241	4.31 × 10^4^	4.53	3.38 × 10^5^	100.00	AC090529.2	clone RP31-160L19
Gene_991	5:100650478-100688178	1405	1.36 × 10^4^	−2.56	1.26 × 10^6^	100.00	AC087262.2	clone RP31-151E23
Gene_539	2:205516170-205525297	614	2.60 × 10^5^	9.34	3.05 × 10^7^	100.00	AC079389.2	clone RP31-263K14
Gene_363	1:236356294-236466993	6485	7.56 × 10^5^	12.31	2.87 × 10^7^	100.00	AC079378.2	clone RP31-7L11
Gene_424	2:33937181-33983278	1391	1.89 × 10^5^	−31.70	1.14 × 10^42^	100.00	AC079378.2	clone RP31-7L11
Gene_424	2:33935270-33983278	1481	9.16 × 10^4^	3.33	1.18 × 10^42^	100.00	AC079378.2	clone RP31-7L11
Gene_754	3:154870205-154872439	1160	3.20 × 10^8^	−6.07	5.44 × 10^44^	100.00	AC079378.2	clone RP31-7L11
Gene_2229	12:40266483-40332612	3832	2.48 × 10^4^	−21.32	3.50 × 10^44^	100.00	AB218617.1	clone CH230-65K18
Gene_2487	15:39865097-39873520	897	2.51 × 10^5^	−6.52	7.76 × 10^13^	100.00	AC095845.8	CH230-10C24
Gene_2224	12:38974186-38975238	787	4.17 × 10^5^	8.86	9.47 × 10^14^	100.00	AC097039.8	BAC CH230-61E1
Gene_4449	16:83872552-83875193	2070	1.68 × 10^5^	17.28	5.27 × 10^5^	100.00	AC095195.6	BAC CH230-5J23
Gene_3993	10:10510094-10515742	2519	5.79 × 10^6^	16.55	6.77 × 10^7^	100.00	AC132013.4	BAC CH230-269G5
Gene_3993	10:10510094-10515742	5648	7.01 × 10^4^	21.75	6.77 × 10^7^	100.00	AC132013.4	BAC CH230-269G5
Gene_2043	10:108207735-108209413	1678	9.77 × 10^6^	4.16	2.57 × 10^6^	100.00	AC128611.4	BAC CH230-249K23
Gene_88	1:81050735-81058656	339	2.99 × 10^5^	−28.24	9.51 × 10^7^	100.00	AC095281.6	BAC CH230-10M16
Gene_1807	10:16140312-16148385	2296	2.30 × 10^20^	32.72	1.62 × 10^4^	100.00	AC094950.6	BAC CH230-6H12
Gene_3837	8:61949091-61960895	645	3.15 × 10^4^	8.02	1.04 × 10^42^	100.00	AC091537.2	clone RP31-78C13
Gene_2257	12:52289596-52307976	3292	2.35 × 10^5^	3.29	2.21 × 10^6^	100.00	AC091000.2	clone RP31-485F9
Gene_1975	10:88721220-88731424	915	2.13 × 10^8^	15.04	5.51 × 10^30^	100.00	AC091514.2	clone RP31-223K12
Gene_2044	10:108197514-108207915	8978	2.35 × 10^7^	3.87	3.45 × 10^12^	100.00	AC087722.2	clone RP31-198L13
Gene_4266	14:5550182-5556214	424	1.32 × 10^4^	2.64	5.59 × 10^8^	100.00	AC087722.2	clone RP31-198L13
Gene_2137	12:992649-1008813	4578	3.66 × 10^4^	−7.57	5.41 × 10^7^	100.00	AC087722.2	clone RP31-198L13
Gene_2137	12:992649-1008813	4317	8.03 × 10^4^	−6.96	5.41 × 10^7^	100.00	AC087722.2	clone RP31-198L13
Gene_1808	10:16149065-16154830	5344	3.31 × 10^17^	10.22	3.22 × 10^5^	100.00	AC090529.2	clone RP31-160L19
Gene_1808	10:16149065-16154830	5765	9.16 × 10^15^	13.41	3.22 × 10^5^	100.00	AC090529.2	clone RP31-160L19
Gene_38	1:52785873-52810395	5598	9.61 × 10^15^	8.83	2.28 × 10^7^	100.00	AC087262.2	clone RP31-151E23
Gene_1582	8:104634314-104656536	961	8.57 × 10^5^	6.79	2.07 × 10^7^	100.00	AC087262.2	clone RP31-151E23
Gene_3812	8:33136091-33153195	7456	1.48 × 10^4^	12.63	1.59 × 10^7^	100.00	AC087262.2	clone RP31-151E23
Gene_2354	14:5744018-5746923	298	4.82 × 10^4^	−2.66	7.45 × 10^9^	100.00	AC087262.2	clone RP31-151E23
Gene_86	1:80618017-80630218	1402	5.20 × 10^4^	2.63	3.15 × 10^8^	100.00	AC087262.2	clone RP31-151E23
Gene_3977	10:16154557-16162465	7908	4.19 × 10^15^	15.41	2.04 × 10^8^	100.00	AC079389.2	clone RP31-263K14
Gene_1626	8:129175363-129180711	3264	9.80 × 10^11^	12.50	4.96 × 10^8^	100.00	AC079389.2	clone RP31-263K14
Gene_1752	9:113942674-113948528	576	1.84 × 10^5^	9.05	1.87 × 10^37^	100.00	AC079389.2	clone RP31-263K14
Gene_1753	9:113952791-114001447	817	1.14 × 10^9^	8.60	1.60 × 10^21^	100.00	AC080157.26	RP32-475K22
Gene_172	1:137413500-137427306	2703	6.43 × 10^4^	−2.01	2.15 × 10^5^	100.00	AC105654.4	BAC CH230-201P14
Gene_4216	13:6344942-6347579	1405	6.94 × 10^5^	6.02	3.98 × 10^21^	100.00	AJ297736.1	hsp86
Gene_2541	16:7199973-7203706	3733	1.76 × 10^4^	−10.96	7.47 × 10^5^	100.00	AC111654.6	BAC CH230-108G17
Gene_2086	11:39085999-39093587	7533	4.30 × 10^4^	−3.06	5.48 × 10^4^	100.00	AC095195.6	BAC CH230-5J23
Gene_3947	10:10509959-10516032	5836	4.23 × 10^7^	73.20	7.28 × 10^7^	100.00	AC132013.4	BAC CH230-269G5
Gene_3947	10:10509959-10516032	3181	4.60 × 10^7^	26.59	7.28 × 10^7^	100.00	AC132013.4	BAC CH230-269G5
Gene_3947	10:10509959-10516032	5512	5.63 × 10^6^	24.26	7.28 × 10^7^	100.00	AC132013.4	BAC CH230-269G5
Gene_3947	10:10509959-10516032	6073	4.05 × 10^5^	59.93	7.28 × 10^7^	100.00	AC132013.4	BAC CH230-269G5
Gene_3947	10:10509959-10516032	5749	7.53 × 10^4^	27.74	7.28 × 10^7^	100.00	AC132013.4	BAC CH230-269G5
Gene_2043	10:108207735-108209413	1678	1.84 × 10^17^	8.43	2.57 × 10^6^	100.00	AC128611.4	BAC CH230-249K23
Gene_1617	8:122685629-122689781	417	9.80 × 10^4^	−17.87	2.99 × 10^4^	100.00	AC120734.5	BAC CH230-220D1
Gene_846	4:61876180-61892396	10229	4.68 × 10^8^	5.65	1.51 × 10^7^	100.00	AC095876.6	BAC CH230-10G12
Gene_846	4:61876180-61892397	10230	1.58 × 10^7^	5.68	1.51 × 10^7^	100.00	AC095876.6	BAC CH230-10G12
Gene_846	4:61876180-61892396	14726	2.31 × 10^5^	7.13	1.51 × 10^7^	100.00	AC095876.6	BAC CH230-10G12
Gene_846	4:61876180-61892397	15413	5.10 × 10^4^	3.36	1.51 × 10^7^	100.00	AC095876.6	BAC CH230-10G12
Gene_1807	10:16140312-16148385	2296	3.86 × 10^41^	47.78	1.62 × 10^4^	100.00	AC094950.6	BAC CH230-6H12
Gene_1807	10:16132421-16148466	16045	9.80 × 10^4^	16.99	3.23 × 10^4^	100.00	AC094950.6	BAC CH230-6H12

Notes: FC = fold change; Identify = the highest percent identity for a set of aligned segments to the same subject sequence; Pred. gene accession = accession number of predicted genes in the NCBI database. The blue, orange, and green backgrounds mean results generated in the CT vs. H30, CT vs. H60, and CT vs. H120 comparisons.

**Table 5 biology-11-01740-t005:** Summary of differential isoform-corresponding genes that were also differentially expressed (DEIDEGs).

Comparisons	No. DEIDEGs	No. Up-Regulated DEIDEGs	No. Down-Regulated DEIDEGs
B_CT vs. H120	43	25	18
L_CT vs. H30	81	15	66
L_CT vs. H60	452	187	265
L_CT vs. H120	253	134	119
A_CT vs. H30	132	107	25
A_CT vs. H60	451	353	98
A_CT vs. H120	566	405	161

Note: DEIDEGs refer to the differential isoform corresponding genes that were also differentially expressed. B, L, and A mean blood, liver, and adrenal glands.

**Table 6 biology-11-01740-t006:** Summary of the DEIDEGs enriched in processes of response to heat- and cold-induced thermogenesis under heat stress.

Comparisons	Gene Name	Position	FC	FDR
Response to heat L_CT vs. H60	Bag3	1:199941160-199965191	13.18	1.13 × 10^16^
	Hsp90aa1	6:135107270-135112775	3.73	1.13 × 10^8^
	Dnaja1	5:57028466-57039378	2.96	1.23 × 10^8^
	Hmox1	19:14508615-14515456	3.28	1.85 × 10^7^
	Dnaja4	8:59278261-59294003	7.24	4.61 × 10^7^
	Pklr	2:188449209-188459592	−3.56	1.27 × 10^5^
	Chordc1	8:17421556-17446165	2.53	1.02 × 10^3^
	Cdkn1a	20:6351457-6358864	9.40	1.45 × 10^3^
	LOC680121	11:13499163-13501263	2.22	1.53 × 10^3^
	Hspa1b	20:4877323-4879779	1281.27	1.05 × 10^59^
	Tp53inp1	5:24410862-24416888	11.16	2.54 × 10^4^
	LOC103694877	AABR07024106.1:11533-12195	−2.32	2.99 × 10^3^
	Mif	20:13732197-13732859	−2.34	3.12 × 10^3^
	Atp2a2	12:39553902-39603326	2.27	5.63 × 10^3^
Response to heat L_CT vs. H120	Hsp90aa1	6:135107270-135112775	3.84	2.99 × 10^10^
	Pklr	2:188449209-188459592	−3.06	3.85 × 10^9^
	AABR07011951.1	2:177651240-177653288	3.74	1.61 × 10^8^
	Abcc2	1:263554452-263613252	2.28	4.77 × 10^8^
	Hmox1	19:14508615-14515456	3.61	6.85 × 10^6^
	Dnaja4	8:59278261-59294003	8.29	9.12 × 10^6^
	Bag3	1:199941160-199965191	5.56	3.04 × 10^4^
	Dnaja1	5:57028466-57039378	2.35	3.40 × 10^4^
	Cdkn1a	20:6351457-6358864	5.82	1.51 × 10^3^
	Chordc1	8:17421556-17446165	2.45	2.00 × 10^3^
Cold-induced thermogenesis L_CT vs. H120	Fabp5	2:93981655-93985378	−6.24	6.83 × 10^14^
	Prkab1	12:46316235-46326790	−3.17	9.30 × 10^9^
	Lpin1	6:41799748-41870046	4.09	4.31 × 10^4^
	Scd	1:264160128-264172729	−3.07	2.11 × 10^3^
Response to heat A_CT vs. H30	Hspa1b	20:4877323-4879779	35.00	1.84 × 10^13^
	Bag3	1:199941160-199965191	4.20	7.50 × 10^6^
Response to heat A_CT vs. H60	Bag3	1:199941160-199965191	35.79	2.05 × 10^47^
	Hspa1b	20:4877323-4879779	182.12	8.50 × 10^38^
	Dnaja1	5:57028466-57039378	9.43	6.77 × 10^32^
	Hsp90aa1	6:135107270-135112775	9.56	2.95 × 10^30^
	LOC680121	11:13499163-13501263	5.44	4.21 × 10^27^
	Dnaja4	8:59278261-59294003	16.65	8.66 × 10^23^
	Chordc1	8:17421556-17446165	5.15	7.58 × 10^14^
	AABR07011951.1	2:177651240-177653288	3.68	2.71 × 10^11^
	Hmox1	19:14508615-14515456	5.29	1.57 × 10^9^
	Hspd1	9:61680529-61690956	2.02	1.72 × 10^8^
	Cdkn1a	20:6351457-6358864	4.73	1.95 × 10^4^
	Hspa5	3:13838303-13842762	2.05	6.98 × 10^5^
Response to heat A_CT vs. H120	LOC680121	11:13499163-13501263	7.37	1.50 × 10^53^
	Hsp90aa1	6:135107270-135112775	11.35	2.46 × 10^37^
	Dnaja1	5:57028466-57039378	9.24	1.55 × 10^25^
	Dnaja4	8:59278261-59294003	22.41	3.09 × 10^23^
	AABR07011951.1	2:177651240-177653288	4.01	8.87 × 10^18^
	Chordc1	8:17421556-17446165	5.06	1.14 × 10^15^
	Vcp	5:58426548-58445953	2.12	8.30 × 10^13^
	Bag3	1:199941160-199965191	13.50	2.75 × 10^11^
	Hspd1	9:61680529-61690956	2.04	4.29 × 10^10^
	Hmox1	19:14508615-14515456	5.19	2.18 × 10^8^
	Cdkn1a	20:6351457-6358864	5.95	2.58 × 10^6^
	Trpv2	10:48903539-48925030	−3.02	2.33 × 10^5^
Cold-induced thermogenesis A_CT vs. H30	AABR07033324.1	11:17336443-17340373	3.41	3.09 × 10^3^
	Gadd45g	17:13391466-13393243	5.60	6.99 × 10^7^
	Arntl	1:178039062-178137465	7.45	2.22 × 10^6^
	Gnas	3:172374956-172428483	4.59	4.77 × 10^6^
	Cebpb	3:164424514-164425910	2.48	9.99 × 10^5^
Cold-induced thermogenesis A_CT vs. H60	Atf4	7:121480722-121482772	3.16	1.50 × 10^12^
	Arntl	1:178039062-178137465	9.73	5.39 × 10^8^
	Gnas	3:172374956-172428483	4.02	6.04 × 10^8^
	Gadd45g	17:13391466-13393243	7.05	9.29 × 10^8^
	Acsl1	16:48937455-49003246	−2.09	7.51 × 10^3^
	Igf1r	1:128924965-129206516	2.76	2.00 × 10^2^
Cold-induced thermogenesis A_CT vs. H120	Arntl	1:178039062-178137465	17.47	1.72 × 10^21^
	Stk11	7:12440750-12457513	3.06	9.21 × 10^15^
	Atf4	7:121480722-121482772	3.56	1.21 × 10^3^
	Vegfa	9:17340340-17355681	2.19	2.37 × 10^13^
	Gadd45g	17:13391466-13393243	7.37	4.47 × 10^12^
	Trpv2	10:48903539-48925030	−3.02	2.33 × 10^5^
	Dync1h1	6:134958853-135085769	2.05	1.10 × 10^3^
	Acsl1	16:48937455-49003246	−2.11	1.33 × 10^3^
	Igf1r	1:128924965-129206516	3.16	1.67 × 10^3^

Note: FC means fold change, FDR means the false discovery rate, and L and A mean the liver and adrenal glands.

## Data Availability

The RNA-Sequencing datasets of the CT and H120 groups (blood, liver, and adrenal glands) and datasets of the H30 and H60 groups (liver and adrenal glands) used in this work are available in the Sequence Read Archive (SRA) at the National Center for Biotechnology Information (NCBI) with BioProject IDs PRJNA589869 and PRJNA690189, respectively.

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
