# Peer review of "Identification of Novel mRNA Isoforms Associated with Acute Heat Stress Response Using RNA Sequencing Data in Sprague Dawley Rats"

_biology, 2022, doi:10.3390/biology11121740_

Round 1

Author Response

Please see the file attached

Reviewer 2 Report

The authors carried out a trial on the gene expression profile of rats housed at different temperatures.

My comments and suggestions on the submitted manuscript:

- Authors mention in the summary, introduction sections that they had a previous study with the same treatments on rats to identify differentially expressed genes using RNA-seq. It is not clear what the new scientific activity is in the present manuscript: a new animal experiment? new RNA seq? or same database?

- Please describe the method of analysis as samples from two of the treatments (control group and H120 group) were sequenced by HiSeq2000 and the rest (H30 and H60) using HiSeq2500 equipment. (Eg. assay, chemicals, same lab or different lab)

- Fig 1: in the legend describe the meaning of CT, H30, H60 and H120.

- Use the word "feed" for animals.

- What is the reason for the different number of samples collected (4 for blood vs. 5 for liver/adrenal gland)?

What could be the reason that there are no shared annotated DEIs among CT-H30, CT-H60 versus CT-H120 in adrenal glands? (Figure 2)

Author Response

Please see the file attached

Reviewer 3 Report

Review reports

A brief summary

This study is a good opportunity to investigate the genetic makeup for animals’ responses to heat stress and detection for the changes that occur during the trial of animals to cope with such stress. This study needs further work on how to benefit from those identified genes not only in the selection of heat tolerant animals to breed in such hot areas in the world but also in the prevention of heat stress through detecting the time and degree of heat stress which lead to these changes in the gene’s expression. The manuscript needs more clarification that it depends on a previous study in which the heat stress was applied to animals and samples collected.

Specific comments 

Summary:

-        Line 16-17: add (the genetic makeup of) before (heat stress)

-        Line 19-21: write (the present) instead of (this), add (aim was the)before (identification), add (which) before (narrow), add (that) before (underlying), write (underly) not (underlying), remove (regulatory)

-        Line 22-24: write (these) instead of (the), remove (found in this study)

Abstract:

-        Line 26-27: the aim from the current study should be mentioned at the start of the abstract before going on the details. Write (on) not (with), add (three heat stressed groups housed at) before (420c)

-        Line 34: add (also these) after (and), remove (the)

-        Line 36: write (about) instead of (a total of)

-        Line 41: write (with) instead of (to improve)

Introduction:

-        Line 50: write (adverse effect) not (impairment)

-        Line 51: add (there is limited information about) before (the molecular), remove (are not clear)

-        Line 53-54: remove (with), don`t repeat the word (response) and let be at the end (responses)

-        Line 60-62: add (was found to) before (plays), write (play) not (plays), add (by reference 14) before (has shown)

-        Line 63: add (was found to be involved) before (in the process)

-        Line 64-66: rewrite to: reported that an alternatively spliced mRNA with 169 additional nucleotides in the 5’ noncoding region was found after heat shock in comparison with the mRNA transcribed under non-heat shock conditions

-        Line 74-78: too long statement please shorten it into smaller statements

-        Line 79: add (number of mRNA isoforms involved in) before (expression)

-        Line 80: write (under different) not (and) before (heat stress)

-        Line 81: write (the) before (novel)

Materials and Methods:

-        Line 88-89: add (were) before (purchased), write (to be used) instead of (were used)

-        Line 91: need to mention briefly how heat stress conditions was established even if it is done in previous study

-        Line 92: why not collecting blood from the H30 and H60 as control and H120?

-        Line 91-94: too long please divide it into smaller statements

-        Line 94: write (in which) instead of (and) before (the blood)

-        Line 99: you need to mention is the blood collected in a tube containing or not anticoagulant, what do you mean by the white film layer of blood?

-        Line 100: remove (stored) before (in a 2 ml)

-        Line 106: reagent not regent

-        Line 159: as not an

-        Line 153: stages not groups, experiment not experiments

Results and Discussion:

-        Line 195-199: should be added at the start of material and methods not here

-        Line 206: you didn`t mention the replicates in the methods

-        I didn`t find figure S1A and S1B

-        Line 214: lowerly not lowly

-        Line 222: lowerly not lowly

-        Line 271: in figure 2 mention the meaning of circles color

-        Line 280: studies not study, add (the) before (metabolism)

-        Line 303: add (the) before (response)

-        Line 415: add (the) before (process)

Conclusion:

-        Line 500: add (were) before (involved)

-        Line 502: add (which) before (could)

-        Line 506: move the word (mammals) after (select)

Author Response

Please see the file attached
